# Prognostic Significance of Prostate-Specific Antigen Persistence after Radical Prostatectomy: A Systematic Review and Meta-Analysis

**DOI:** 10.3390/cancers13050948

**Published:** 2021-02-24

**Authors:** Shoji Kimura, Fumihiko Urabe, Hiroshi Sasaki, Takahiro Kimura, Kenta Miki, Shin Egawa

**Affiliations:** Department of Urology, Jikei University School of Medicine, Tokyo 105-8461, Japan; shoji.kimura.0221@gmail.com (S.K.); furabe0809@gmail.com (F.U.); shiroshi427@gmail.com (H.S.); kentamiki@gmail.com (K.M.); s-egpro@jikei.ac.jp (S.E.)

**Keywords:** PSA persistence, nodal involvement, prostate cancer, recurrence, meta-analysis

## Abstract

**Simple Summary:**

Radical prostatectomy, the standard treatment for localized or locally advanced prostate cancer, generally provides good disease control and favorable long-term survival. However, about 40% of patients experience biochemical recurrence, and a significant number of them develop clinical progression. If identified in advance, those patients could receive specially tailored follow-up and counseling. Current guidelines recommend the first prostate-specific antigen tests three months after surgery, but prostate-specific antigen levels should be undetectable within four weeks. Early testing at 4–8 weeks after surgery could thus be useful for determining patient prognosis. We reviewed the medical literature to study persistence of prostate-specific antigens, progression of prostate cancer, and survival of patients. We found that prostate cancer was more likely to recur in patients with or without lymph nodal involvement who had prostate-specific antigen persistence at 4–8 weeks.

**Abstract:**

We performed a systematic review and meta-analysis to assess the prognostic value of prostate-specific antigen (PSA) persistence 4–8 weeks after radical prostatectomy (RP) in patients with prostate cancer, using studies from Medline, Scopus, and Cochrane Library, on 10 October 2020. Studies were eligible if they compared patients with postoperative PSA persistence 4–8 weeks after RP to those without such persistence to assess the value of PSA persistence in prognosticating biochemical recurrence (BCR), disease recurrence, cancer-specific mortality (CSM), and overall mortality (OM) by multivariable analysis. Our review and analysis included nine studies published between 2008 and 2019 with 14,455 patients. Of those studies, 12.0% showed postoperative PSA persistence. PSA persistence was associated with BCR (HR: 4.44, 95% CI: 2.84–6.93), disease recurrence (HR: 3.43, 95% CI: 1.62–7.25), and CSM (HR: 2.32, 95% CI: 1.83–2.95). We omitted meta-analysis on the association of PSA persistence with OM due to an insufficient number of studies. PSA persistence was associated with disease recurrence in a sub-group of patients with pathological nodal involvement (HR: 5.90, 95% CI: 3.76–9.24). Understanding detection of PSA persistence at 4–8 weeks after RP might be useful for patient counseling, follow-up scheduling, and clinical decision-making regarding adjuvant therapies.

## 1. Introduction

In 2020, it was estimated that 191,930 patients were diagnosed with prostate cancer (PCa), and 33,330 died of that disease in the US. Additionally, PCa is the second leading cause of cancer-related death in men in the US alone [1] and is also increasing in its incidence and mortality in the world. Most cases initially diagnosed are localized to the prostate and found in men between 60 and 70 years old with life expectancy more than 15 years. Radical prostatectomy (RP) with or without extended lymph node dissection is one of the standard treatments for patients diagnosed with localized and locally advanced PCa, and generally provides local disease control and long-term survival [2]. Recently, the improvement of surgical techniques such as laparoscopic surgery and robotic surgery has been proposed. These techniques are expected to improve patients’ quality of life and oncologic and survival outcomes after RP [3]. However, approximately 40% of patients treated with RP experience biochemical recurrence (BCR), with a significant proportion experiencing clinical disease progression within a decade after BCR diagnosis on average [4]. Identifying patients at increased risk of BCR would allow for tailoring of evidence-based follow-up intensity and could improve counseling and decision-making regarding adjuvant or early salvage therapy [5].

PSA measurement after surgery is the established follow-up tool for patients treated with RP. The detectable PSA after surgery indicates either residual PCa, residual benign tissue, recurrence in the prostatic bed or distant micro-metastases, or a combination of both.

In fact, the EAU-ESTRO-SIOG guideline recommends the first PSA measurement at three months after RP [3]. However, considering the half-life of PSA, undetectable PSA levels would be expected within four weeks after RP [6]. Therefore, detectable PSA at 4–8 weeks after RP could be a relevant prognostic factor and might help identify the patients likely to benefit from intensified follow-up scheduling and adjuvant therapy.

Nasseli et al. studied PCa treated with RP and found an association between PSA persistence (>0.1 ng/mL) at six weeks after RP and lower BCR-free survival rates. Additionally, some studies reported an association between PSA persistence (>0.1 ng/mL) at six weeks after RP and lower disease recurrence-free survival rates in the patients with lymph nodal involvement at RP. Recently, other papers have shown PSA persistence (>0.1 ng/mL) at 4–8 weeks after RP to also predict disease recurrence, cancer-specific mortality (CSM), and overall mortality (OM). We hypothesized that PSA persistence (>0.1 mg/mL) at 4–8 weeks would influence these oncologic and survival outcomes in PCa patients treated with RP and there are no systematic reviews and meta-analyses on this topic at this moment. In this study, therefore, we performed a systematic review and meta-analysis to investigate the association of PSA persistence (>0.1 ng/mL) at 4–8 weeks after RP with oncologic and survival outcomes.

## 2. Results

### 2.1. Study Selection and Characteristics

Overall, 698 articles were identified for initial assessment and 5 additional articles were identified from reference lists of original articles and review articles (Figure 1): out of these, 160 duplicates were removed, 421 articles were excluded after assessing the title and abstract, and an additional 108 articles were excluded after reading the full text. At the end of this process, nine studies remained that investigated the impact of PSA persistence on oncologic outcomes in 14,455 patients treated with RP. Those nine studies were included in this systematic review and meta-analysis [7,8,9,10,11,12,13,14,15].

The general characteristics of the eligible studies are summarized in Table 1. All studies had a retrospective design and were published between 2008 and 2019. Of these, six studies enrolled patients from Europe, two from Northern America, and one from Asia. The patients in this systematic review and meta-analysis were treated with RP between 1990 and 2017. Operation modality varied among included studies, such as open, laparoscopic, and robotic surgery. Four studies did not report about the operation modality. Seven studies reported the association between disease recurrence and PSA persistence. Three and two studies reported the association of PSA persistence with CSM and BCR, respectively. Only one study reported the association of OM. Most included studies had intermediate and high quality of study based on The Newcastle–Ottawa Scale. Clinicopathological characteristics are summarized in Table 2. The median age of included patients was between 60 and 67. The median initial PSA varied among studies. Overall, 1697 of 14,172 patients (12.0%) treated with RP experienced PSA persistence 4–8 weeks postoperatively. Advanced disease (≥pT3a) was reported in 5484 of 14,455 patients (37.9%). A positive surgical margin was reported in 3426 of 14,455 patients (23.7%). Lymph node involvement was reported in 1395 of 14,339 patients (9.7%).

We did not perform a meta-analysis on the association of PSA persistence with OM, as only one study showed an association between these two variables. Preisser et al. showed a significant impact of PSA persistence on OM (HR: 1.86, 95% CI: 1.41–2.45) in a multivariable Cox regression analysis [11].

No studies were rated as high risk of bias in the overall rating. We judged five studies to have moderate risk of bias and four studies to have low risk of bias in the overall rating (Appendix A).

### 2.2. Meta-Analysis

#### 2.2.1. Association between PSA Persistence and BCR

The impact of PSA persistence on BCR was reported in two studies, including 434 patients treated with RP. Forest plots (Figure 2A) showed that PSA persistence was significantly associated with BCR (pooled HR: 4.44; 95% CI, 2.84–6.93; z = 6.54). The Cochrane Q test (Chi^2^ = 0.01; *p* = 0.926) and I^2^ test (I2 = 0.0%) did not show significant heterogeneity. Funnel plots did not identify any studies over the pseudo 95% CI (Figure 2A).

#### 2.2.2. Association between PSA Persistence and Disease Recurrence

The impact of PSA persistence on disease recurrence was reported in seven studies, including 14,021 patients treated with RP. The forest plots (Figure 2B) showed that PSA persistence was significantly associated with disease recurrence (pooled HR: 3.43; 95% CI, 1.62–7.25; z = 3.22). The Cochrane Q test (Chi^2^ = 149.15; *p* < 0.01) and I^2^ test (I^2^ = 96.0%) showed significant heterogeneity. The funnel plots identified four studies over the pseudo 95% CI (Figure 2B).

#### 2.2.3. Association between PSA Persistence and CSM

The impact of PSA persistence on CSM was reported in three studies, including 12,356 patients treated with RP. Forest plots (Figure 2C) showed that PSA persistence was significantly associated with CSM (pooled HR: 2.32; 95% CI, 1.83–2.95; z = 6.90). The Cochrane Q test (Chi^2^ = 2.21; *p* = 0.331) and I^2^ test (I^2^ = 9.5%) did not show significant heterogeneity. Funnel plots did not identify any studies over the pseudo 95% CI (Figure 2C).

#### 2.2.4. Subgroup Analysis

The impact of PSA persistence on disease recurrence was assessed in a sub-group of patients with pathological nodal involvement in RP specimens. Meta-analysis was not performed for other outcomes, such as BCR, CSM, or OM, due to an insufficient number of studies for assessment. Three studies enrolling 582 patients were identified. Forest plots (Figure 3) showed that PSA persistence was significantly associated with disease recurrence (pooled HR: 5.90; 95% CI: 3.76–9.24; z = 7.74). The Cochrane Q test (Chi^2^ = 3.93; *p* = 0.140) and I^2^ test (I2 = 49.1%) did not show significant heterogeneity. Funnel plots did not show any studies over the pseudo 95% CI (Figure 3).

## 3. Discussion

In this systematic review and meta-analysis, we analyzed nine studies comprising a total of 14,455 patients treated with RP for PCa. As far as we know, this is the first system-atic review and meta-analysis to investigate the impact of PSA persistence four to eight weeks after RP on oncologic and survival outcomes, such as BCR, disease recurrence, and CSM. As only one study evaluated the association of PSA persistence with OM, we did not perform a meta-analysis for OM after RP. In this systematic review and meta-analysis, approximately 12% of patients treated with RP experienced PSA persistence postopera-tively. We found that PSA persistence was associated with BCR, disease recurrence, and CSM after RP. Additionally, we found that PSA persistence was associated with disease recurrence in patients diagnosed with pathological nodal involvement in RP specimens.

In this systematic review and meta-analysis, we analyzed nine studies comprising a total of 14,455 patients treated with RP for PCa. As far as we know, this is the first systematic review and meta-analysis to investigate the impact of PSA persistence four to eight weeks after RP on oncologic and survival outcomes, such as BCR, disease recurrence, and CSM. As only one study evaluated the association of PSA persistence with OM, we did not perform a meta-analysis for OM after RP. In this systematic review and meta-analysis, approximately 12% of patients treated with RP experienced PSA persistence postoperatively. We found that PSA persistence was associated with BCR, disease recurrence, and CSM after RP. Additionally, we found that PSA persistence was associated with disease recurrence in patients diagnosed with pathological nodal involvement in RP specimens.

PSA measurement after surgery is the established follow-up tool for patients treated with RP. Since the PSA half-life is 3.15 days [16], the PSA value is expected to reach an undetectable level within four weeks in patients who have undergone complete pathological resection. As a result, persistently detectable PSA levels between four and eight weeks after RP indicate either residual PCa, residual benign tissue, recurrence in the prostatic bed or distant micro-metastasis, or a combination of both.

Meanwhile, approximately 10% of clinically localized PCa with RP and extended pelvic lymph node dissection show lymph nodal involvement at the final pathologic analysis [17,18]. Historically, node-positive patients were considered to be affected by systemic disease [19]. However, recent studies have demonstrated that the outcomes of surgically treated patients with lymph nodal involvement are not invariably poor [20,21,22]. In addition, we still lack evidence on adjuvant strategy in patients with pathological lymph nodal involvement. Therefore, individualized PCa management requires accurate identification of patients with lymph nodal involvement at higher risk of disease recurrence after RP.

Several nomograms have been validated for predicting BCR or CSM after RP comprising standard clinicopathologic features. These predictive findings in each nomogram guide physicians and patients into adjuvant therapy or intensified follow-up scheduling after RP. However, some nomograms’ accuracy is around 0.80 of the concordance index, which means moderate predictivity [23,24]. For example, 10–20% of patients with favorable pathological features, such as low Gleason score, negative surgical margin, and lymph node involvement will experience BCR [25,26]. To improve accuracy, reliable validated markers need to be integrated into conventional nomograms.

Our results highlight that PSA persistence at 4–8 weeks may identify patients with an increased risk of BCR, disease recurrence, and CSM. Therefore, we hypothesize that the PSA persistence may improve current available prognostic models and help physicians in their clinical decision-making regarding adjuvant strategies such as radiation therapy after RP for PCa patients and patient counseling follow-up scheduling. Additionally, some patients with pathological lymph node involvement might benefit from adjuvant therapy, and some could avoid unnecessary adjuvant therapy if they are found to be in a category where less recurrence is expected.

Despite showing a strong correlation between PSA persistence and oncologic outcomes, this meta-analysis has several limitations. First, we included only retrospective cohort studies, and patients were excluded from the analysis if complete information was not available, which may have created a selection bias. Only a few studies investigated the impact of PSA persistence on survival outcomes such as CSM and OM, probably due to the short duration of follow-up. The definition of disease recurrence in each study was metastasis on imaging and same among studies. However, the term or expression of that varied among studies, such as “disease metastasis on imaging” and “clinical progression confirmed by imaging”. PSA cut-off used in the definition of BCR varied in each study. Therefore, the result of this meta-analysis regarding BCR should be interpreted with caution. There was a considerable inter-study difference in factors such as the surgeon’s experience with the procedure and the surgical methods used, which may have introduced a confounder. Individual pathologists described pathological results independently, and no study provided a central pathological assessment. Heterogeneity was detected in the meta-analysis of disease recurrence, limiting the value of these results. Finally, the studies included in this systematic review defined PSA persistence as detectable, more than 0.1 ng/mL, and did not calculate the optimal cut-off level for PSA persistence. Despite these limitations, we confirmed the impact of PSA persistence on oncologic outcomes in patients who have undergone RP for PCa. PSA measurement is easily accessible and noninvasive, and early measurement could be integrated easily into clinical care. Well-designed prospective studies with longer follow-up are needed in the future to add further evidence on the prognostic impact of PSA persistence and to define optimal cut-off values.

## 4. Materials and Methods

This systematic review and meta-analysis was performed following the Preferred Reporting Items for Systematic Reviews and Meta-Analyses (PRISMA) statement [27] and the Cochrane Handbook for Systematic Reviews of Interventions [28]. The protocol in this systematic review and meta-analysis has been registered in the International Prospective Register of Systematic Reviews database (PROSPERO: CRD 42020219515) and is available in full on the University of York website. On 10 October, 2020, we conducted a comprehensive literature search using electronic databases (MEDLINE, Scopus, and Cochrane Library) to retrieve published articles assessing the association of PSA persistence (>0.1 ng/mL) at 4–8 weeks after RP with oncologic and survival outcomes. All full-text papers were assessed. Papers were excluded if they were deemed inappropriate after screening the title and abstract. The process was conducted independently by 2 reviewers (S.K. and F.U.), with disagreements resolved by the senior author (T.K.). The following MESH terms were used: ((prostate cancer) AND (((prostatectomy) OR (surgery)) OR (surgical))) AND ((PSA persistence) OR (PSA persistent)).

The studies were considered eligible if PCa patients (localized or locally advanced disease) who had undergone RP (Population) with PSA persistence (0.1 ng/mL) at 4–8 weeks after surgery (Exposure) were compared to PCa patients (localized or locally advanced disease) without PSA persistence (0.1 ng/mL) at 4–8 weeks after surgery (Comparator). This process identified independent predictors of oncologic and survival outcomes (Outcome) using the multivariate Cox regression analysis in non-randomized observational or cohort studies (Study design). BCR, disease recurrence, CSM, and OM were our primary outcomes of interest. Studies evaluating the prognostic impact of PSA persistence at more than 3 months after RP or using a cut-off value for PSA persistence of more than 0.2 ng/mL were excluded. Papers were also excluded if they were review articles, editorials, commentaries, documents published in a language other than English, meeting abstracts, replies from authors, or case reports. If multiple articles had been written by the same group based on similar patient cohorts, only the most extensive or most recently published article was included.

Two authors (S.K. and F.U.), working independently, extracted the required data from all eligible studies. Baseline clinicopathological characteristics were collected, and the hazard ratios (HRs) and 95% confidence intervals (CIs) were calculated for PSA persistence associated with each of the outcomes. All discrepancies regarding data extraction were resolved by consensus with the assistance of the third investigator (T.K.).

Because the studies to be included were observational, we extracted adjusted HRs or ORs and 95% confidence intervals to calculate the cumulative effect size. Studies with Kaplan–Meier log-rank tests, univariable Cox proportional hazard regression, or logistic regression analyses were not considered for the meta-analysis. To avoid introducing a high level of additional selection bias, we did not use effect summary estimation methods. Statistical pooling of effect measures was based on the level of heterogeneity among studies as assessed by the Cochrane Q test and I2 statistics. As indicated by a *p*-value < 0.05 in Cochrane Q tests and a ratio >50% in I2 statistics, significant heterogeneity led to the use of random-effect models following the DerSimonian and Laird method. If these tests were negative for heterogeneity, we calculated pooled HRs by applying the inverse-variance method [29,30,31]. We evaluated for publication bias, including the small-study effect, by visually inspecting funnel plots for all assessed comparisons. Statistical analyses were performed using STATA/MP 14.2 (Stata Corp., College Station, TX, USA).

We used the Quality in Prognosis Studies (QUIPS) tool [32] to assess the risk of bias. Each study was assessed for risk of bias through six domains: study participation, study attrition, prognostic factor measurement, outcome measurement, study confounding, statistical analysis and reporting, and overall rating.

For each domain, two review authors (S.K., F.U.) independently assigned a rating of low, moderate, or high risk of bias. Disagreements were resolved by consensus or consultation with the third investigator (T.K.). The risk of bias assessment is reported in Appendix A.

We used The Newcastle–Ottawa Scale to assess the quality of the included studies, based on the Cochrane Handbook for systematic reviews [27,28,33]. The scale focuses on three factors: Selection (1–4), Comparability (1–2), and Exposure (1–3). The total scale ranges from 0 to 9. The main confounders were identified as the important prognosis factors of each oncologic and survival outcome. The presence of confounders was determined by consensus and review of the literature. We identified those studies with scores higher than 6 as “high-quality” choices.

## 5. Conclusions

We performed a systematic review and meta-analysis to assess the prognostic value of PSA persistence 4–8 weeks after RP in patients with prostate cancer. We confirmed that PSA persistence is associated with oncologic outcomes such as BCR, disease recurrence, and CSM in PCa patients after RP, including patients with lymph nodal involvement. Detection of PSA persistence at 4–8 weeks after RP could be useful for patient counseling, follow-up scheduling, and clinical decision-making regarding adjuvant therapies.

## Figures and Tables

**Figure 1 cancers-13-00948-f001:**
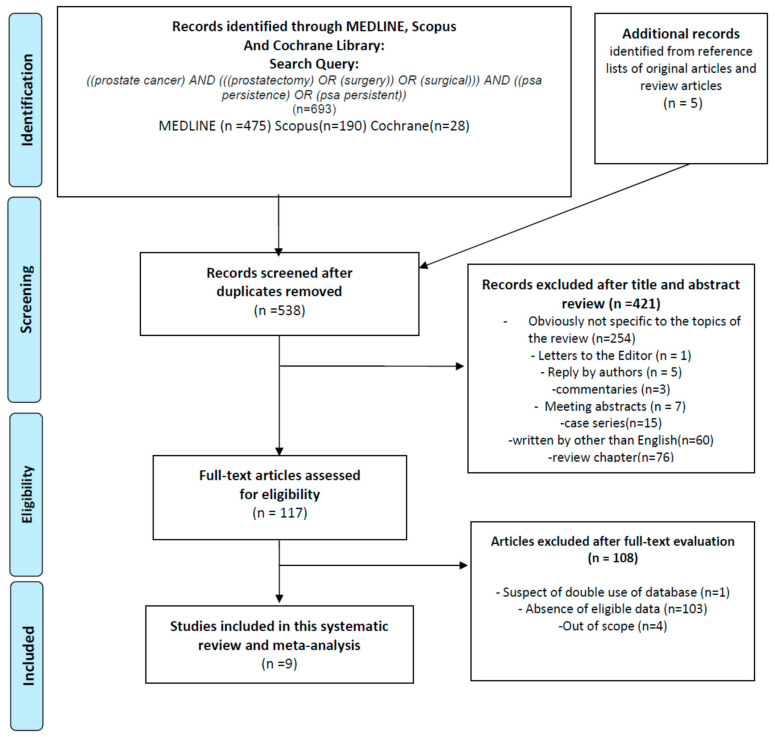
Flow chart for article selection process to analyze the prognostic significance of PSA persistence after radical prostatectomy in patients with prostate cancer.

**Figure 2 cancers-13-00948-f002:**
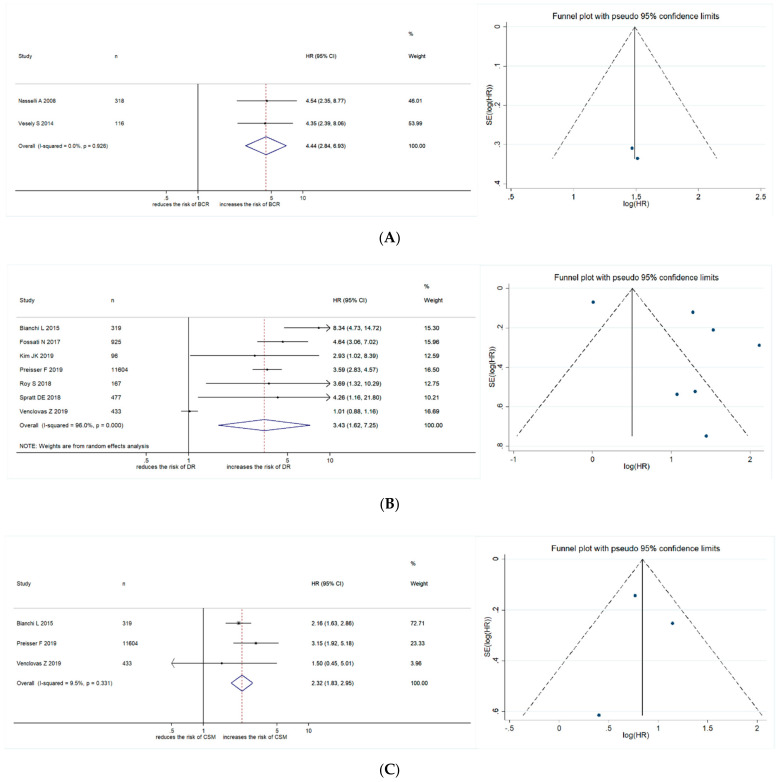
Forest plots and funnel plots showing the association of PSA persistence with (**A**) biochemical recurrence, (**B**) disease recurrence and (**C**) cancer-specific mortality in Prostate Cancer patients treated with radical prostatectomy.

**Figure 3 cancers-13-00948-f003:**
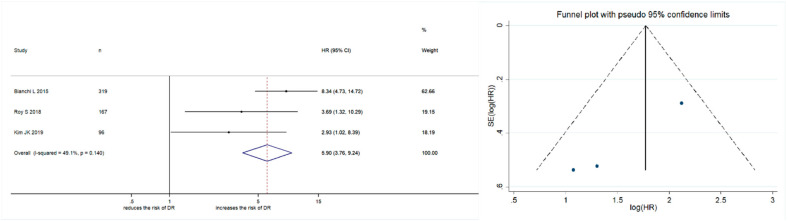
Forest plot and funnel plot showing the association of PSA persistence with disease recurrence in prostate cancer patients with pathological lymph nodal involvement at radical prostatectomy.

**Table 1 cancers-13-00948-t001:** Characteristics of eligible studies included in the systematic review.

Author and Year	Region	Design	Recruitment Period	No. pts	Operation Modality	Reported Outcomes	NOS	Funding Source
Nasselli A 2008	Italy	R cohort	2002–2007	318	ORP	BCR	6	NR
Vesely S 2014	Czech	R case-control	2001–2012	116	ORP or LRP	BCR	5	NR
Bianchi L 2015	Italy	R cohort	1998–2013	319	NR	Disease recurrence, CSM	7	None
Fossati N 2017	EU	R cohort	1996–2009	925	NR	Disease recurrence	7	None
Roy S 2018	Canada	R cohort e	2005–2014	167	NR	Disease recurrence	6	Sanofi and Bayer Health Care Pharmaceuticals
Spratt DE 2018	US	R cohort	1990–2015	477	NR	Disease recurrence	6	GenomeDx Biosciences
Kim JK 2019	Korea	R cohort	2002–2014	96	ORP or RARP	Disease recurrence	6	The National Research Foundation of Korea
Preisser F 2019	Germany	R cohort	1992–2016	11,604	ORP or RARP	Disease recurrence, CSM, OM	7	None
Venclovas Z 2019	Lithuania	R case-control	2001–2017	433	ORP	Disease recurrence, CSM	5	NR

BCR: biochemical recurrence, CSM: cancer-specific free mortality, LRP: laparoscopic radical prostatectomy, NOS: The Newcastle-Ottawa Scale, NR: not reported, ORP: open radical prostatectomy, OM: overall mortality, R: retrospective, RARP: robot-assisted radical prostatectomy, US: United States.

**Table 2 cancers-13-00948-t002:** Patient characteristics of nine studies included in this systematic review.

Author and Year	Median F/U (Overall or Undetectable vs. Persistence)	Median age (Overall or Undetectable vs. Persistence)	Median iPSA (IQR) <Overall or Undetectable vs. Persistence>	GS ≥ 8	≥pT3a	PSM	pN+	*n*. PSA Persistence	*n*. Postoperative RT
Nasselli A 2008	NR	65	7 (NR)	77(24.2%) (GS ≥ 4 + 3)	79(24.8%)	89(28.0%)	20(6.3%)	33(10.4%)	NR
Vesely S 2014	31.4 months	64	9.2(2.9–38.2)	59(51%) (GS ≥ 3 + 4)	62(53.4%)	116(100%)	NR	NR	NR
Bianchi L 2015	53 months	65	11.1(7–23.3)	156(52.5%)	278(87.1%)	169(53.0%)	319(100%)	83(26%)	ART 200(62.7%)
Fossati N 2017	8 years	65	8.0(5.6–13.5)	228(24%)	519(56%)	403(44%)	0	224 (24.2%)	925(100%)
Roy S 2018	48 months	64	12.5(8.2–21.5)	102(61.3%)	156(94%)	117(70%)	167(100%)	NR	63(37.7%)
Spratt DE 2018	57 months	60	6.4 (NR)	NR	242(51.6%)	229(48.0%)	38(8.0%)	150(31.4%)	NR
Kim JK 2019	45 months	67	30.5 (NR) vs. 30.6 (NR)	39(40.6%)	89(92.7%)	73(76%)	96(100%)	52(54.2%)	ART 20(20.8%)
Preisser F 2019	61.8 months vs. 46.4 months	64.6 vs. 64.2	6.6 (4.7–9.7) vs. 11.2 (6.8–19.8)	NR	3764(32.5%)	2042(17.6%)	699(6.0%)	1025(8.8%)	1815(15.6%)
Venclovas Z 2019	64 months	65	NR	189(43.6%)	295(68.1%)	188(43.4%)	56(12.9%)	130(30%)	NR

ART: adjuvant radiation therapy, F/U: follow-up, GS: Gleason score, NR: not reported, PSM: positive surgical margin, RT: radiation therapy.

## Data Availability

No new data were created or analyzed in this study. Data sharing is not applicable to this article.

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
