# Peer review of "Prognostic Significance of Prostate-Specific Antigen Persistence after Radical Prostatectomy: A Systematic Review and Meta-Analysis"

_cancers, 2021, doi:10.3390/cancers13050948_

Round 1
Reviewer 1 Report
Authors performed systematic review and meta-analysis regarding PSA persistence and some kinds of survival. They logically concluded that PSA persistence is associated with oncologic outcomes such as biochemical recurrence, disease recurrence and cancer-specific mortality. The theme of this article is very interesting and important for clinicians performing radical prostatectomy (RP) for patients with local or locally advanced prostate cancer. After RP, confirming pathological diagnosis, clinicians and patients is carefully observing serum PSA level. This systematic review and meta-analysis can help them to decide adjuvant therapy, salvage radiation therapy or something else, with enough evidences. Thus, this article is worth publish without any point to be revised.
Author Response
Authors performed systematic review and meta-analysis regarding PSA persistence and some kinds of survival. They logically concluded that PSA persistence is associated with oncologic outcomes such as biochemical recurrence, disease recurrence and cancer-specific mortality. The theme of this article is very interesting and important for clinicians performing radical prostatectomy (RP) for patients with local or locally advanced prostate cancer. After RP, confirming pathological diagnosis, clinicians and patients is carefully observing serum PSA level. This systematic review and meta-analysis can help them to decide adjuvant therapy, salvage radiation therapy or something else, with enough evidences. Thus, this article is worth publish without any point to be revised.
Reply)Thank you very much for his/her comments.
Reviewer 2 Report
General comments
In the systematic review and meta-analysis, the prognostic value of PSA persistence 4-8 weeks after radical prostatectomy in patients with prostate cancer was assessed. The present manuscript was well designed as systematic review and meta-analysis, and written for the Journal of “Cancers”. Since the prediction of the biochemical recurrence in early-term follow-up would contribute the tailor-made treatment for the patients with PC, the results of the present review and meta-analysis is very interested, and might be useful for patient counseling, follow-up scheduling, and clinical decision-making regarding adjuvant therapies.
Specific comments
Major criticism
- The present manuscript was well designed as systematic review and meta-analysis, and written for the Journal of “Cancers”. The clinical importance of the present study is significance.
Minor criticism
- “4. Materials and Methods” was located between “3. Discussion” and “5. Conclusions”. Is the order correct?
Author Response
In the systematic review and meta-analysis, the prognostic value of PSA persistence 4-8 weeks after radical prostatectomy in patients with prostate cancer was assessed. The present manuscript was well designed as systematic review and meta-analysis, and written for the Journal of “Cancers”. Since the prediction of the biochemical recurrence in early-term follow-up would contribute the tailor-made treatment for the patients with PC, the results of the present review and meta-analysis is very interested, and might be useful for patient counseling, follow-up scheduling, and clinical decision-making regarding adjuvant therapies.
Specific comments
Major criticism
- The present manuscript was well designed as systematic review and meta-analysis, and written for the Journal of “Cancers”. The clinical importance of the present study is significance.
Reply)Thank you very much for his/her comments.
Minor criticism
- “4. Materials and Methods” was located between “3. Discussion” and “5. Conclusions”. Is the order correct?
Reply)Thank you very much for his/her comments. The order is correct, according to the author's guideline.
Reviewer 3 Report
The meta-analysis is methodologically correctly performed even if, unfortunetly, not enought data are available on OM, data regarding other end points come from retrospective studies and it will not be easy to obtain results with prospective-randomized studies.
Only one point must be reviewed: Introduction-lines 45-46: laparoscopic and robotic tecniques do not improve outcomes compared to open surgery with strong evidence...probably did you mean surgery compared to no active treatment ?
Author Response
The meta-analysis is methodologically correctly performed even if, unfortunetly, not enought data are available on OM, data regarding other end points come from retrospective studies and it will not be easy to obtain results with prospective-randomized studies.
Only one point must be reviewed: Introduction-lines 45-46: laparoscopic and robotic tecniques do not improve outcomes compared to open surgery with strong evidence...probably did you mean surgery compared to no active treatment ?
Reply)Thank you very much for his/her comments. There is no strong evidence for improving outcomes in patients treated with minimally invasive surgery. We changed this sentence to the following: “These techniques are expected to improve patients’ quality of life and oncologic and survival outcomes after RP.” (line45-46)
Reviewer 4 Report
I would like to thank the authors for providing this review on a very interesting topic.
Some additional recommendations should be considered before publishing:
Introduction:
- Please provide a reference for the first sentence. The numbers used, are they worldwide of specifically for a country/region? Please specify.
- Line 41-42: radical prostatectomy is not THE standard treatment. EBRT is equally suitable for this patient population (cfr EAU guidelines). please change! Also, if radical prostatectomy is chosen, this is most likely in the context of multimodality treatment (which is more than only RP + PLND). This should me mentioned.
- Could you provide evidence for the statement that laparoscopic and robotic radical prostatectomy improve oncological outcomes compared to open RP?
Also reference no.3 is an old one. Please change to the updated version of the guidelines (2020). - Line 54: not if it is correct to state that detectable PSA after RP is due to recurrence. PCa is a slow growing tumor, we do not expect it to recur within 3 months after radical surgery. I would remove it.
- Line 68: remove “on”
Material and methods:
- Please specify the search period (start date included) + rationale for this period.
- Could you please specify the rationale for including only NRS, both comparative and single arm? I do agree for inclusion of single arm, cohort studies. As this is in fact a prognostic (rather than interventional) SR, single arm studies reporting MVA with persisting PSA as one of the variables, could provide essential information.
- Study population: any PCa, localized or also locally advanced? Please specify.
- Please state more clearly if abstract screening AND full text (selected based on the abstract screening) was performed in duplicate by independent reviewers and how conflicts were resolved.
- My main concern is the used RoB strategy. The authors state that they used the Cochrane Handbook for Systematic Reviews of Interventions to assess the risk-of-bias evaluation of the included studies. Although correct for an interventional SR, this is not one. In my opinion, the authors try to investigate the PROGNOSTIC/PREDICTIVE effect of PSA persistence on different oncological outcomes (as clearly stated in the first sentence of the conclusion). For prognostic systematic reviews, QUIPS tool should be used. For the correctness of this review this needs to be adapted.
- Please provide funding information of each study included in this systematic review in the summary tables.
- A suggestion: if possible provide a summary figure of the RoB analysis, where you can show the overall bias per domain. Very informative to see which biases are present and in what extend
Results:
- Concerning article inclusion LINE 76, what do you mean with “and from further 76 up-to-date literature search “
- Please report RoB analysis/results more elaborative as this is important for data interpretation
- Line 94-95: I don’t understand how a median PSA of 31 can be explained by merely lymph node involvement. I would remove this.
- Patient characteristics table: who did you use cut-offs for GS, T stage? Is it clinical T3a or pathological T-stage
- Please state in the table the study type of each paper.
- Please provide IQR after median PSA in the tables.
- Why did you perform a meta-analysis on the subgroup of patients with lymph node involvement?
- As you clearly state that definitions of disease recurrence significantly varied amongst studies included, could you explain to me why a meta-analysis was possible? First of all, meta-analysis of non-randomised controlled trials should be avoided. However, if the outcomes reported in non-randomised studies are sufficiently similarly defined, it may be appropriate and feasible to display the results of these studies in a forest plot, of course omitting a pooled estimate of effect. Therefore, I would reconsult with a statistician about the validity of the performed analyses. The same statements account for the other two (CSS and BCR). How was BCR defined in the included papers? Moreover, omitting a pooled estimate of effect should be considered as all studies were non-randomised controlled trials.
Author Response
I would like to thank the authors for providing this review on a very interesting topic.
Some additional recommendations should be considered before publishing:
Introduction:
- Please provide a reference for the first sentence. The numbers used, are they worldwide of specifically for a country/region? Please specify.
Reply)Thank you very much for his/her comments. The number was that of the US. We added in line37 the following sentence: in the US.
- Line 41-42: radical prostatectomy is not THE standard treatment. EBRT is equally suitable for this patient population (cfr EAU guidelines). please change! Also, if radical prostatectomy is chosen, this is most likely in the context of multimodality treatment (which is more than only RP + PLND). This should me mentioned.
Reply)Thank you very much for his/her comments. We agree to this comment. RP is one of the standard treatments same as EBRT. We changed this sentence following: Radical prostatectomy (RP) with or without extended lymph node dissection is one of the standard treatments for patients diagnosed with localized and locally advanced PCa, and…….(line41-43)
- Could you provide evidence for the statement that laparoscopic and robotic radical prostatectomy improve oncological outcomes compared to open RP?
Reply)Thank you very much for his/her comments. There is no strong evidence for improving outcomes in patients treated with minimally invasive surgery. We changed this sentence following: “These techniques are expected to improve patients’ quality of life and oncologic and survival outcomes after RP.” (line45-46)
Also reference no.3 is an old one. Please change to the updated version of the guidelines (2020).
Reply)Thank you very much for his/her comments. We changed to the updated version of the guidelines.
- Line 54: not if it is correct to state that detectable PSA after RP is due to recurrence. PCa is a slow growing tumor, we do not expect it to recur within 3 months after radical surgery. I would remove it.
Reply)Thank you very much for his/her comments. We mean detectable PSA after surgery in this sentence is not specific “within 3 months”. Generally, the detectable PSA after surgery (at any time point) would indicate either residual PCa, residual benign tissue, recurrence in the prostatic bed or distant micro-metastases, or a combination of both.
- Line 68: remove “on”
Reply)Thank you very much for his/her comments. We removed “on”
Material and methods:
- Please specify the search period (start date included) + rationale for this period.
Reply)Thank you very much for his/her comments. We searched without limit of search period not to miss the literatures which have the possibility of eligibility.
- Could you please specify the rationale for including only NRS, both comparative and single arm? I do agree for inclusion of single arm, cohort studies. As this is in fact a prognostic (rather than interventional) SR, single arm studies reporting MVA with persisting PSA as one of the variables, could provide essential information.
Reply)Thank you very much for his/her comments. We agree reviewer’s comments. We are thinking RCT is not suitable for MA of prognostic factor, because of the short follow-up duration in RCT (generally speaking). Therefore, we decided to include only NRS, both comparative and single arm.
Study population: any PCa, localized or also locally advanced? Please specify.
Reply)Thank you very much for his/her comments. This SR and MA included non-metastatic PCa patients, which means localized or locally advanced disease. We changed in line 224 and 227.
- Please state more clearly if abstract screening AND full text (selected based on the abstract screening) was performed in duplicate by independent reviewers and how conflicts were resolved.
Reply)Thank you very much for his/her comments. Shoji Kimura and Fumihiko Urabe performed this process. If there are conflicts in this process, senior author (Takahiro Kimura) made a decision. We changed in line 220-221.
- My main concern is the used RoB strategy. The authors state that they used the Cochrane Handbook for Systematic Reviews of Interventions to assess the risk-of-bias evaluation of the included studies. Although correct for an interventional SR, this is not one. In my opinion, the authors try to investigate the PROGNOSTIC/PREDICTIVE effect of PSA persistence on different oncological outcomes (as clearly stated in the first sentence of the conclusion). For prognostic systematic reviews, QUIPS tool should be used. For the correctness of this review this needs to be adapted.
Reply)Thank you very much for his/her comments. We agree reviewer’s comment. We change ROBINS to QUIPS tool in assessing the RoB strategy. In materials and method section, we changed to the following: We used the Quality in Prognosis Studies (QUIPS) tool [32] to assess the risk of bias. Each study was assessed for risk of bias through six domains: study participation, study attrition, prognostic factor measurement, outcome measurement, study confounding, statistical analysis and reporting, and overall rating.
For each domain, two review authors (S.K, F.U) independently assigned a rating of low, moderate, or high risk of bias. Disagreements were resolved by consensus or consultation with the third investigator (T.K). The risk of bias assessment is reported in Table S1. (line256-263)
- Please provide funding information of each study included in this systematic review in the summary tables.
Reply)Thank you very much for his/her comments. We added funding information for each study in Table1.
- A suggestion: if possible provide a summary figure of the RoB analysis, where you can show the overall bias per domain. Very informative to see which biases are present and in what extend
Reply)Thank you very much for his/her comments. We changed to Table S1 about the QUIPS tool which can be shown the overall bias per domain.
Results:
- Concerning article inclusion à LINE 76, what do you mean with “and from further 76 up-to-date literature search “
Reply)Thank you very much for his/her comments. We did not perform further up-to-date literature search in the article selection process. Sorry for the confusion. We changed in line 77 and Figure1.
- Please report RoB analysis/results more elaborative as this is important for data interpretation
Reply)Thank you very much for his/her comments. We showed RoB results in line 102-104 and TableS1
- Line 94-95: I don’t understand how a median PSA of 31 can be explained by merely lymph node involvement. I would remove this.
Reply)Thank you very much for his/her comments. We agree to this comment. We removed this sentence.
- Patient characteristics table: who did you use cut-offs for GS, T stage? Is it clinical T3a or pathological T-stage
Reply)Thank you very much for his/her comments. Patients characteristics are varied among included studies. Some studies reported these cut-offs. We used these cut-offs to make readers understand visibly easily and to clarify the rate of high GS or locally advanced disease.
T stage is the pathological stage. We changed in Table2.
Please state in the table the study type of each paper.
Reply)Thank you very much for his/her comments. We added the study type of each study in Table1.
- Please provide IQR after median PSA in the tables.
Reply)Thank you very much for his/her comments. We added IQR of median PSA of each study in Table2.
- Why did you perform a meta-analysis on the subgroup of patients with lymph node involvement?
Reply)Thank you very much for his/her comments. At this moment, there is a lack of evidence for adjuvant therapy in patients with pathological N+. We are looking for the predictive factor that might identify the patients with pathological N+ who are likely to have disease recurrence and benefit from adjuvant therapy. Therefore, we performed a meta-analysis on the subgroup of patients with lymph node involvement and found PSA persistent was associated with disease recurrence in patients with lymph node involvement.
- As you clearly state that definitions of disease recurrence significantly varied amongst studies included, could you explain to me why a meta-analysis was possible? First of all, meta-analysis of non-randomised controlled trials should be avoided. However, if the outcomes reported in non-randomised studies are sufficiently similarly defined, it may be appropriate and feasible to display the results of these studies in a forest plot, of course omitting a pooled estimate of effect. Therefore, I would reconsult with a statistician about the validity of the performed analyses. The same statements account for the other two (CSS and BCR). How was BCR defined in the included papers? Moreover, omitting a pooled estimate of effect should be considered as all studies were non-randomised controlled trials.
Reply)Thank you very much for his/her comments. We defined disease recurrence in this systematic review and meta-analysis as metastasis on imaging. but the term of “disease recurrence” varied among studies, such as “metastasis on imaging” and “clinical progression confirmed by imaging” et al. For example, Venclovas et al. defined CP as skeletal or visceral confirmed by bone scan, computer tomography (CT), positron emission tomography (PET/CT), or magnetic resonance imaging (MRI) in their study. Therefore, the interpretation of disease recurrence is almost the same among studies but varied in the term or expressions in each paper. Sorry for the confusion and making mistakes. We changed in Discussion part the following: The definition of disease recurrence in each study was metastasis on imaging and almost the same among studies. However, the term or expression of that varied among studies, such as "disease metastasis on imaging" and "clinical progression confirmed by imaging". (line192-195)
Second, PSA cut-off of BCR in two studies varied (0.2 and 0.4). However, the PSA cut-off of BCR post nadir depends on time trend and clinician’s discretion in clinical practice. Therefore, we are thinking this discrimination is not influential but should interpret this result with caution. We added in the limitation part of the Discussion the following: PSA cut-off using in the definition of BCR in each study varied. Therefore, the result of meta-analysis regarding BCR should be interpreted with caution. (line195-197)
Finally, there is no clear description and eligibility in PRISMA regarding whether or not combine data of non-randomized studies. At this time, our team decided to perform meta-analyses in this systematic review.
Thank you very much again for your meaningful comments which could improve our manuscript.
Round 2
Reviewer 4 Report
I have no further comment. The changes significantly improved the paper.